# How to Write a Positivist Legal History: Lessons from the 18th and 19th Centuries English Jurists William Blackstone and James Fitzjames Stephen

Susanna Menis 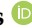

School of Law, Birkbeck, University of London, Malet Street, London WC1E 7HX, UK; s.menis@bbk.ac.uk

**Abstract:** This paper is about the shaping of the law understood as a positivist enterprise. Positivist law has been the object of contentious debate. Since the 1960s, and with the surfacing of revisionist histories, it has been suggested that the abstraction of the doctrine of criminal law is due to its categorisation in early histories. However, it is argued here that positivism was hardly an intentional master plan of autocratic social control. Rather, it is important to recognise that historians do not provide a value-free recount of history. This paper examines this assertion by drawing on the writings of the English jurists William Blackstone and his work *Commentaries on the Law of England* (1765), and James Fitzjames Stephen's *A History of the Criminal Law of England* (1883). Taking these scholars not as mere a-historical writers but reflecting on the fact that they inevitably 'functioned' as conduits of their own social practise opens an inquiry into the social response to a social need, which was already under way long before their time.

**Keywords:** positivist law; histories of law; Blackstone; J.F. Stephen; criminal law

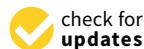



## 1. Introduction

At the core of this study are two jurists well known by English scholars. Although one is 100 years older than the other, and some would consider it inconceivable that they should be part of the same story—indeed, in this study they are, not least because they were part of a tradition of a way of seeing things. Both William Blackstone and James Fitzjames Stephen have been presented as the fathers of criminal law pedagogy and doctrine [1] (p. 3). After all, textbook structure and discourse concerning the teaching of the discipline still follow, to a great extent, what Blackstone proposed in his *Commentaries on the Law of England* (1765), and what Stephen also later expounded in his digest *A History of the Criminal Law of England* (1883).

Blackstone was a scholar, a lawyer, a judge, and a member of the House of Lords. He formalised the legal training he started at the Inns of Middle Temple in 1741 with a Bachelor of Civil (Roman) Law at Oxford University in 1745 [2]. Douglas explains that he 'divided his time between the university and the Inn of Court; for in the former, he attached himself to his academic pursuits, and in the latter, applied himself closely to his profession' [3] (p. 7). Blackstone progressed to become a university lecturer at Oxford's All-Souls College; in 1758, having been elected as the first Vernerian Professor of the Common Law of England, he had the first chance to introduce his *Commentaries*. In later years and centuries, the *Commentaries* attracted a wide range of critiques; however, its importance at the time it was published (in several volumes) is summarised by Douglas as the 'most happy circumstance to the profession as well as to the lecturer' because it provided 'one of the most valuable forensic productions ever published in this, or perhaps, in any other country' [2,3].

James Fitzjames Stephen never met Blackstone; still, he became engaged with Blackstone's original aim of creating a systematic, clear, and coherent criminal law digest to be used by students and lawyers. Stephen was initially better known for his journalistic

legal writings; his brother Leslie Stephen explains that in his contributions to the *Saturday Review,* Stephen 'denounce(d) the cumbrous and perplexed state of the law in general so energetically' [4] (p. 154). In 1869, he was appointed legal member of the Indian viceroy's council [4] (pp. 231–236); for about three years, India became Stephen's playground for drafting legislative reforms. Upon returning to England, he attempted to apply this expertise by proposing several criminal law-related bills, yet with little success [5] (more about Stephen's time in India and his attempted codifications in England in [4] (pp. 237–291, 351–381). Stephen's biggest piece of writing was completed during his judgeship in the High Court. Stephen's *History* (1883) has been criticized for its lack of accuracy and depth; however, reviewing the work, Sir F. Pollock suggested that it 'is the most extensive and arduous undertaken by any English lawyer since the days of Blackstone' (Cited in [4] (p. 418).

In their political lives, both men were prominent figures, social activists in their own way and involved in promoting legislation, which was meant to strengthen and improve the socio-political framework of their country. However, this paper does not concern itself with Blackstone and Stephen as public and historical figures; rather, the aim is to explore why the teaching of criminal law took the form of a positivist enterprise: that is, law which has been posited (laid down) by the legal system of the country. This is in contradiction to the natural law theory, for example, which recognises the legitimacy of a law in ways which are more profound than the mere cohesive power of the state.

Blackstone and Stephen championed the need for a rationalised and coherent literal presentation of the legal system; however, their efforts have become the object of contentious debate. Since the 1960s, and with the surfacing of revisionist histories (See explanation in [6] (p. 220). For an example of such history see [7]), it has been suggested that the 'pains' caused by the abstraction of the doctrine of criminal law are due to its categorisation in early histories such as those by Blackstone (The writing mainly refers to [8]) and Stephen (The writing mainly refers to [9]), so much so that some critical legal scholars have gone as far as arguing that these histories meant to disguise the friction between the state and the people, 'an attempt to mystify both dominator and dominated by convincing them of the "naturalness", the "freedom", and the "rationality" of a condition of bondage' [10] (pp. 209–210). However, despite the valuable contribution of critical legal scholarship, and particularly from the perspective of power relations, it is argued here that this theoretical framework gives too much credit to the idea of unified/premeditated intentional actions and thoughts by the 'dominator'. Blackstone and Stephen were deeply involved in public life, in academia, the press, the government, and the judiciary; however, to argue that their views represented or were shared by a majority of others could be misleading. Historical records suggest that they maintained independence of thought: Blackstone was a Tory sympathiser within a Whig government [11], whilst Stephen had some reservations about Benthamite liberalism [12,13]. Although liberal political ideologies were emerging during their lifetimes, both experienced the rejection of their proposals for penal and legal reform.

The following is what Robert Gordon has termed external legal history (Cited in [14] (p. 168). The overall aim of this history is to explore how the teaching of criminal law (and perhaps law in general) developed to become positivist in nature; that is, it is analytically oriented to facilitate a breaking down of the criminal law doctrine into components which are a-human, homogeneous, value-neutral, and abstract in their nature. This historical examination is significant because, although we recognise that the common law is a slowly evolving, breathing entity driven by humans for humans, and as such it is at the mercy of human whim, histories of the common law have been taken as a-historical. In other words, histories of law have been read at face value; as if, in their interpretation, nothing but predominantly power-relation interests have been taken as responsible for the driving and shaping of the law. The paper takes Blackstone and Stephen as the departing point for examination. It takes the two main documents that they wrote, as best put by Arnold, as 'truth' beyond 'what actually happened': this approach can 'demonstrate *how* people think,

the images and language and associations they can have upon their culture' [15] (p. 9 [italic in original]).

Indeed, the English common law (whether statute or case law) is a man-made law, capriciously affected by those potent forces of human nature. Gearey et al. further argue that common law is deceptively coherent; its entity 'is deeply penetrated by historical legal experience' [16] (p. 59). Furthermore, Hespanha explains that 'the mission of legal history is to render problematic the implicit assumptions of dogmatics, namely, the rational, necessary, ultimate nature of our laws' [17] (p. 41). Here, however, the assumptions challenged are the dogmas constructed by legal revisionist histories in their critique of traditional histories. It is argued here that too much attention has been placed by revisionists on what traditional histories said, rather than on why they said what they said. This, in itself, has reproduced exactly what they meant to criticise; that is, a positivist formulation of the law. Indeed, these critical histories tend to focus on the socio-legal role of criminal law, penal institutions, or the philosophical understanding of legal responsibility as expressed through traditional histories (See for example: [18,19]). By following this line of enquiry, however, even critical and socio-legal scholars inevitably find themselves trapped in the positivist net. In his examination of 'proof' and 'truth', Ginzburg explains that the 'sceptical approach' (of anti-positivism) to historical evidence 'precludes any access to reality'—and this, according to Ginzburg, 'turns out to be a sort of inverted positivism' [20] (p. 84). Therefore, assessing histories such as those by Blackstone and Stephen as more than mere legal writings can provide some important considerations on the question of the shaping of modern criminal law; taking these scholars not as mere a-historical writers, but reflecting on the fact that they inevitably 'functioned' as conduits of their own social practise, reveals an active social response to a social need, which was already under way long before Blackstone and after Stephen. Indeed Skinner has considered that taking such a contextual examination could make the 'idea' or 'thought' under examination (i.e., legal positivism) lose its 'timeless element' and 'universal application' [21] (p. 4). However, the examination here focuses less on the idea of legal positivism but rather centres on those elements that helped to shape a trend that had been generated out of a way of thinking and doing things (that would later on be labelled as legal positivism).

The paper starts by setting the context of the discussion. First, it briefly addresses scholarly critique of positivist law; it then moves on to consider the legacy left by writings such as the *Commentaries* and *A History of the Criminal Law of England*. Next, the discussion turns to consider the historical writing styles that might have affected the way criminal law was subsequently shaped and taught. The examination places attention on the relationship between positivism, the notion of 'scientific knowledge', and grand narratives. The final part of the paper examines the need, as considered by Blackstone, Stephen, and their contemporaries, to have a systematic university course of legal studies in order to improve the practise of law and to incentivise legal reform.

## 2. Critique and Legacy

Eighteenth and nineteenth century historical writing styles have affected knowledge production up to this day; however, the styles have also attracted severe criticism. Blackstone's style is a product of the Enlightenment or philosophical history, which is typical of the eighteenth century; Stephen's style fits into the period of the mid-nineteenth century, when what was termed as romantic or political history embraced the use of archives and the idea of studying and writing history as a science [22] (pp. 8, 25, 43, 68). Despite spanning two different centuries and presenting different literal characteristics, these styles were later labelled Whig histories (also known as Traditional or Administrative histories) [22] (p. 64). Considered to be exaggerated and a-critical, eighteenth to nineteenth century legal historiography triggered a sense of repulsion among socio-political, legal, and philosophy scholars of the 1960s and onwards [23] (p. 15); [24] (p. 89). The historian Herbert Butterfield had already warned of the 'dangers' of such histories in the 1930s and had discredited their congratulatory, manipulative, and abridged style of writing [25] (p. 5, 11, 17, 24). It is

especially this argument that has led critical legal scholarship to read the stories told by those labelled Whig historians at face value; in other words, anything written before what has been classed as 'social' (legal) histories has been identified, a priori, as Whig, and as such, generally assumed to have little value in its contribution to our knowledge because of its misleading construction of social reality.

Writing in 1910, Arthur Lyon Cross suggested that legal histories, including Blackstone's and Stephen's, contributed little to the study of English law [26] (p. 13). This assertion is debatable, given that there are twentieth and twenty-first century legal scholars still troubled by the huge impact left by these Whig legal writings. The 'contribution' is significant, albeit not necessarily perceived as positive, given that some scholars have attributed the 'pains' of modern criminal law to these writings. In drawing scholars' attention to the system of oppression showcased by the *Commentaries*, Kennedy wrote in 1979 that although Blackstone made many contributions to the utopian enterprise of legality, his *Commentaries* as a whole quite patently attempt to 'naturalize' purely social phenomena. They restate as 'freedom' what we see as servitude [10] (pp. 205, 211).

By establishing the foundations of a positivist view of law, Norrie explains, narratives such as those of Blackstone and Stephen limited the alternative and critical viewpoints on the legitimacy of law. In other words, positivism fostered the assumption that 'the law is a realm of homogeneous concepts that are cut off from the outside world of morality and politics' [7] (p. 87). Critical legal scholars have paid great attention to this issue, expressing concern that textbooks and legal education have fortified and supported 'a pervasive system of oppressive, inegalitarian relations' [27] (p. 1). Indeed, the 1960s and 1970s brought about new social histories, which attempted to 'transform' the writing of the history of crime and punishment (See related discussion in [28]). Influenced by the critical legal tradition and the postmodernist movement, authors started challenging orthodox legal history writing, choosing to address the historical development of criminal law while placing it within a certain theoretical framework—Marxism, power relations and institutional control, class, gender, race, and others [29] (p. 209). For example, Norrie argues that criminal law developed in the way it did because of the eighteenth and nineteenth century middle-classes' concern for the protection of their own property and economic interests [7].

Certainly, this point was evidenced by Austin in his writing: 'the matter of jurisprudence is positive law: law strictly so called, that is, law set by political superiors to political inferiors' [30] (p. 5). Undoubtedly, the power relation expressed by Austin and his contemporaries is problematic and it has been at the core of critical legal analysis; however, taking this statement out of context is also problematic. Indeed, in his critique on the analysis undertaken by the branch of history of ideas, Skinner draws attention to the risks of interpreting a work of literature (as reflective of an idea or a thought) at face value [21] (p. 3). In other words, Skinner suggests that interpreting an author's idea by drawing on one's own modern understanding of the thought can in itself lead to a deceptive construction of knowledge [21] (p. 8). It may well be that Blackstone and certainly Stephen were exposed at some point to Bentham's, Austin's, and others' evolving notion of legal positivism [31] (p. 722) (However, what is known as 'rational law' has already been formalised in the Netherlands, France, and Germany since the 16th century [32] (p. 6); nonetheless, as Skinner suggests:

> *'Besides this crude possibility of crediting a writer with a meaning he could not have intended to convey, since that meaning was not available to him, there is also the danger of too readily "reading in" a doctrine which a given writer might in principle have meant to state, but in fact had no intention to convey'* [21] (p. 9).

Significantly, systematising and compartmentalising the law was interpreted by contemporary critics as a way—to put it simply—to control the citizen; however, those responsible for this systematisation have suggested, as will be discussed below, that this was done for the ease of use and application of the law. Students of law will notice that criminal law textbooks have two parts, and teaching will normally follow this scheme as well: the

general and the special parts. The title of Farmer's 1996 article *The Obsession with Definition* exemplifies the point made about the fixation on this systematic way of understanding the law. The textbook by Lacey, Well, and Quick, *Reconstructing Criminal Law,* states that 'the idea that there is anything general to be said about "the nature of crime" begins to look very problematic indeed'; and yet the authors confess that they 'remain [ . . . ] deeply unfashionable', and opt to merely 'illuminate rather than replace the traditional approach' in the study of law [33] (p. 4). In the 1902 edition of Courtney Stanhope Kenny's *Outlines of Criminal Law*, the analysis of the criminal law doctrine is already based upon William Blackstone's, Stephen's, and Frederic Maitland's texts on criminal law. Kenny's textbook was then modernised, although the structure of the syllabus was maintained, by one of the most prominent English legal scholars, William Glanville, in 1953. Since then, criminal law textbooks, whether traditional, critical, or socio-legal, have followed a notably similar format.

Blackstone's and Stephen's writings have done much to shape the understanding of how the law should be studied and applied. However, their legacy goes beyond that; we only need to skim through the many new editions of these texts that are periodically produced. Indeed, the value of what are claimed to be first editions is huge; recently, such a copy was sold for USD 14,500 (Sold by Raptis Rare Books on February 2019 [34]). To commemorate 250 years of the *Commentaries*, Prest and Widener curated an exhibition that toured the US, London, and Australia in 2015; this included items such as students' notes on the *Commentaries* and diagrams used to accompany the study of the text [35]. Contemporary use of the texts can be further proved by identifying the extent these have been cited in legal decisions. Allen found that the *Commentaries* featured in the US Supreme Court's decisions between 2000 and 2012 in one of every thirteen cases [36] (p. 217). A quick search of the legal database Westlaw.uk reveals 843 cases referring to Blackstone's *Commentaries* from 1765 to 2019, and 155 times in cases from 2000 to 2019. Google Scholar reveals 57,600 hits for 'Blackstone Commentaries'; these includes journal articles about Blackstone's *Commentaries* as well as several new editions of the text. Finalising the search with 'Since 2015', Google Scholar comes up with 8370 hits; while with 'since 2019' (including 2020), records stand on 3740 hits (as for December 2019). A search in the Hein Online database on 'William Blackstone' produces 82,993 results; 'Blackstone's Commentaries' generates 58,780 results–with 1796 results from 2016 to 2019. Although a similar search on J.F. Stephen suggests lesser fame, a Google Scholars search still provides 9600 results for the search term 'James Fitzjames Stephen', with only one for 'A History of the Criminal Law of England', although this has been cited 2174 times (as for December 2019). On Hein Online, 'James Fitzjames Stephen' generates 5982 results, 126 being from 2016 to 2019; searching 'James Fitzjames Stephen A History of the Criminal Law of England' provides 4356 results. Searching Westlaw.uk reveals 53 cases from 1917 to 2017 resulting from 'James Fitzjames Stephen any writing on criminal law', and 12 cases from 1917 to 2008 resulting from the search 'James Fitzjames Stephen A History of the Criminal Law of England'.

## 3. The Science of 'True' Knowledge

Facts, knowledge, and truth have been the object of assiduous debate by historian and legal scholars. Indeed, postmodernism has challenged, since the 1960s, the 'truth' of things; as explained by Donnelly and Norton, 'historical knowledge' is constructed by historians in the act of writing rather than 'being intrinsic to the past itself, waiting to be recovered' [24] (p. 90). Undoubtedly, the 'truth' of positivist law, as it was shaped by the 'scientific knowledge' approach of the eighteenth and nineteenth centuries, reflects a form of knowledge but not necessarily an objective one, nor an exclusive one [24]. However, Blackstone and Stephen wrote in a period when scientific principles enjoyed, as put by Tosh, 'unrivalled prestige' [37] (p. 150). Both were fully engaged in the trend of scientific discourse; hence, despite writing in two different centuries and reflecting different literal styles, their writings are representative of the cumulative nature of the scientific tradition.

Indeed, the 'movement' that was the 'scientific' rationalisation of the law (as many other disciplines, see Headrick 2000) was operating across Europe and beyond. Sugarman, Shapiro, and Masferrer explain that the shift to 'systematisation' and 'generalisation' was driven, in particular, by the seventeenth century intellectual revolution [38] (pp. 105–106); [39] (p. 728); [32] (p. 96). Prest considers that the quest for rationalisation of the common law happened (or started) between the sixteenth and seventeenth centuries [40] (p. 36). Headrick suggests that one of the drivers of scientific development was 'the rising demand for information [ . . . ] stimulated by the growth of population, production, and trade' [41] (p. 11). Berman, in his review on the origins of western legal science, points to its formation at an even earlier period—the eleventh century. He explains that 'legal science was, in the first instance, an institutionalization of the process of resolving social-political conflict by resolving conflicts in authoritative legal texts' [42] (pp. 930–931, 938). However, Masferrer explains that at the end of the eighteenth century and the beginning of the nineteenth century, this trend finally saw its triumph with the codification of the law [32] (p. 96). The Napoleonic Code is a famous example, but other European and non-European countries followed with their own codification: Italy, Spain, Argentina, Brazil, and Canada, just to name a few [32,43,44]. It is certain that England never materialised a codification of its laws; however, it is evident that such first attempt made by Blackstone to formalise the writing of criminal law, and a century later by Steven, to codify it, reflected the need, as in many other countries and as suggested by Masferrer, 'to protect the criminal law from misuse' [43] (p. 7). In England, the relationship between science and positive law is made clear by Amos in his 1872 *The Science of Jurisprudence*:

> *Broadly stated, the Science of Jurisprudence may be described as depending for its materials upon the growth of Positive Law, and the growth of Positive Law as determined by national progress expressed in such facts as improved moral notions and a more highly organised industrial life* [45] (p. 12).

However, according to Sugarman, there was no consistent development in how this 'science' was to be applied [38] (In fact, not all historians view the eighteenth century scientific period positively; Mason judged the first half of the century as 'a singularly bleak period in the history of scientific thought', and Garrison considered the eighteenth century as an 'age of theories and systems (driven by a) mania for sterile, dry-as-dust classifications of everything in nature' (1956 and 1917 respectively, cited in [46] (p. 2). Indeed, Pollock, writing in 1882, suggested that 'not much attention has been paid to the scientific character of the methods by which a great part of English law has actually built up, and by which it is still administrated and developed.' [47] (p. 237). Nevertheless, this scientific approach, albeit unsystematic, led to a 'new' understanding of 'truth' and knowledge. In turn, this brought about a cultural shift, made possible by popularising science (For an interesting analysis, see [48]): writings in print were directed to a popular readership (See analysis in [49]); debates took place in the coffeehouses (Cowen explains that 'by the eighteenth century, the coffeehouse had become a widely accepted part of urban life, and its character as a serious entre for practical learning had been well established' [50] (pp. 99–100); and public lectures offered an intellectual alternative type of leisure (See analysis of public lectures and the public in: [41,51,52]. For public lectures and legal education, see: [53,54]). This, Shapiro explains, 'resulted in a new set of philosophical propositions about the nature of man and his ability to know the world' [39] (p. 732) and [52] (However, scholars have also suggested that too much emphasis has been placed on the 'revolutionary' aspect of science; it has been argued that the process was slow and those elements which enhanced that shift (i.e., universities) were not universally accepted or welcomed [46,55]).

Blackstone's and Stephen's writings are possibly the refined products of at least two centuries of attempts at what could be termed a modern 'scientific' classification and rationalisation of English law. Prest explains that this trend was first expressed in several styles of legal writing. For example, writings such as those by Bacon created collections of rules and maxims; writings by Fraunce, Fulbecke, and others provided techniques for analysis; writings by Finch gave an overview of the whole body of the law [40] (p. 328). The

general aim of these writings, as best put by Prest, was to reduce 'the complex bulk of the law to a neat and comprehensible system' [40] (p. 329). Indeed, in 1600, Fulbecke asserted that 'the common law was a barbarous study, a science void of all proper definitions, artificial divisions, and formal reasons' (Cited in [56] (p. 33). In his historical review of *The Science of Law*, Amos describes the common law prior to the mid-eighteenth century as anomalous, contradictory, barbarous, and irrational, so much so that, according to him, 'it might well be conceived that nothing short of a national cataclysm could suffice to effect a thorough reform of the law without endangering the very basis upon which the whole State rested.' [57] (p. 7).

The scientific trend not only affected writing and education, but for many, it represented a way of life. Similar to the study of law and mingling in the Inns of Court (not necessarily for learning but rather, for socialising) (See [58]. For an earlier period, Prest explains that most of those 'enrolled' in the Inns, whose numbers increased in the seventeenth century, 'sought "social cachet" rather than any sort of legal education' [56] (p. 22), grasping and being engaged with scientific discourses also became part of the intellectual formation of the gentleman. Hale, for instance, wrote several treatises on scientific issues [39] (pp. 740–749); he even 'developed considerable interest and skill' in anatomy and medicine [39] (p. 741). Students and members of the Inns of Court were encouraged to widen their horizons by studying subjects such as anatomy, astronomy, and mathematics [56] (p. 39). They may have left the Inns with little significant knowledge concerning the law, but at least, as Shapiro says, they stayed abreast with fashion, being 'familiar with the substantial scientific activities and literature of the day' [39] (p. 737). Shapiro explains that several judges and lawyers were involved in founding, and were members of, the Royal Society; membership became common in the eighteenth century [39] (p. 738).

Science during Blackstone's and Stephen's period was understood as 'knowledge' and as 'a collection of the general principles or leading truths relating to any subject' [59,60]. Indeed, in his opening chapter 'On the Study of the Law', Blackstone explains that the role of the Vinerian lectureship at Oxford is 'to cultivate and methodize' the 'science thus committed to his charge', which is that of the 'laws and constitution of our own country'; this is essential, according to Blackstone, because it is 'a species of knowledge, in which the gentleman of England have been more remarkably deficient than those of all Europe besides' [8] (p. 4). Blackstone goes on to explain that this is a science 'which distinguishes the criterions of right and wrong; which teaches to establish the one and prevent, punish, or redress the other.' [8] (p. 27). And it is within this context that Blackstone recognises the importance of moving away from a law which is merely natural, because the laws 'denote the rules, not of action in general, but of *human* action or conduct.' [8] (p. 39). Therefore, because 'man was formed for society' [8] (p. 43), these rules have to be 'permanent, uniform, and universal' [8] (p. 44).

Their use of the concept 'science' might have affected Blackstone's and Stephen's writing in dissimilar ways; yet they reflected the same trend aiming to achieve similar outcomes. Indeed, one of the main critiques levelled against Whig histories is that they rationally reconstructed criminal law as a product of 'clear and systematic statements of legal doctrine' [7] (p. 7). Nevertheless, perhaps this was exactly what Blackstone and Stephen set forth to do; that is, create a systematic user-friendly guide, in a period when rules of science acquired a predominant importance in the study and writing of any discipline. Indeed, Meredith, in a letter to Blackstone, expressed his understanding that 'the very end and aim of (your) writings, was to instruct your country-men in the principles of law; not as a dead language, but as a science' [61] (p. 3). Boorstin explains that Blackstone's approach to the writing of the law was affected by the vocabulary used in those days, and therefore he took for 'granted that [ . . . ] the law must be capable of being rationalised and reduced to principles.' [1] (pp. 19, 21). And this was important because, as Amos explains, there was an 'urgent need for such a positive legal system in

order to prevent local order from degenerating into putrid corruption, to give uniformity and consistency to what is heterogenous and scattered [ . . . ]' [45] (p. 13).

Stephen likewise dedicated some thought to the debate concerning the role of science in contemporary society. In fact, he was a member of the Metaphysical Society and a writer of several papers, one of which Gladstone was particularly impressed with: 'Remarks on the Proof of Miracles' [62] (p. 279, note 2). Stephen felt that science and law concerned the 'governing' of human conduct, and therefore these two must be interconnected. This he believed to such an extent that he used the definition of 'law' (referring to Austin's command theory) to explain how science governs human behaviour [63] (pp. 188, 194). In following a scientific approach to the study and application of law, Stephen did not see 'any danger to morality' and thought that 'the interests of morality are, in reality, altogether unaffected' [63] (p. 193). Indeed, the discourse of the 'true' (scientific) legal knowledge is particularly important here. It encouraged the fiction that the development and amelioration of the law had been driven, as Amos assumes, 'by the agency of actual legislation working in obedience to the higher moral aspiration of growing societies' [45] (p. 11). Indeed, it is no surprise that the positivist tradition became the predominant theoretical position; in a lecture at Harvard Law School, Dicey stressed that English law is no mere handicraft or art, but a science to be deduced from a limited number of principles [ . . . ]. Their teaching is scientific because their whole aim is to elucidate the principles of English law [64] (p. 429).

## 4. Grand Narratives

Given that Blackstone's and Stephen's texts have generated much debate as to the validity of their historical truth, it is helpful to examine what the style of historical writing typical of their time meant to achieve. Indeed, both took on a grand project at a time when the quality of legal sources was, as best put by the legal scholar Sir Edward Coke, 'obscure and dark' (Coke 1664, cited in [65] (p. 729). Although Blackstone and Stephen cited major English law histories by scholars such as Bracton, Fortescue, Coke, and Hale, these were at least five centuries, four centuries, and one century old, respectively [26] (p. 7). Moreover, when collating the information, they had to deal with law reports and year books that were unsystematically recorded, hand-written in Latin, and sometimes corrupted; the modern printing of Rolls Series only advanced in the 1860s [66] (pp. 73–75); [67] (p. 206).

A first reading of Blackstone's and Stephen's texts certainly reveals one of the core weaknesses attributed to Whig histories, that is, the sense of a grand narrative and an emphasis on the importance of a sound structural administration of justice. Butterfield identified 'grand' and 'congratulatory' histories as particularly damaging because they only tell a 'great' story, where heroes and villains are portrayed as the moving forces of historical development [25] (p. 63). Written from 'above', these histories tend to narrate the successful story of the current state of affairs; thus, a Whig history of English law will aim at emphasising the greatness of the current system, as opposed and in contrast to previous (or other) inferior jurisdictions. Indeed, Cross criticises Blackstone for being 'a blind admirer of the existing British Constitution' [26] (p. 7); this is evident throughout the *Commentaries*. For example, perhaps one of the most significant congratulatory remarks made by Blackstone is his praise of the British Constitution's ability to embody the three forms of government—democracy, aristocracy, and monarchy—through the separation of powers principle. For him, this is where the 'sovereignty of the British Constitution' and its benefit for society is to be found, 'for in no other shape could we be so certain of finding the three great qualities of government so well and so happily united.' [8] (pp. 48–51).

However, although Blackstone's congratulatory remarks might have been inaccurate, they had a purpose. First, if Blackstone's text is read in the context of eighteenth-century philosophical history, it is not surprising that his style of writing reflected the emphasis on nationalism typical of the time. During this period, scholars were attempting to identify a common ground on which to assert a 'national self-image' [37] (p. 13). Holdsworth explains that the eighteenth century's public figures such as Blackstone 'were no worshippers of

antiquity', and it was 'this intelligent satisfaction with the present [ . . . ] that has given rise to [ . . . ] indiscriminate optimism [ . . . ].' [65] (p. 275). This must have been particularly important for Blackstone, considering that his writing was meant to be studied by law students (or more specifically, the gentleman), that is, possible future societal influencers. In his *Commentaries*, Blackstone attempted to capture on paper a harmonious framework of the English government, which, in reality, may not have been so harmonious after all. In a seventy-page letter to Blackstone, Sheridan accused him of misleading Englishmen's understanding in relation to the 'uncontrolled, absolute, despotic power of Parliament'. 'You have certainly proved yourself every way qualified to be a judge' Sheridan wrote, 'but very unfit to be a legislator'. He went as far as suggesting that Blackstone's description of Parliament was driven by a personal interest: 'you and I Sir William know that the doctrine of the omnipotence of Parliament is a very favourite one in the quarter of promotion [ . . . ]' [68] (pp. 1,4–5). By way of rebutting Blackstone's discussion of the British Constitution, Sheridan provided in his accusatory letter an extensive favourable analysis of Bentham's theory on parliamentary sovereignty. And yet, historical 'truth' may also depend on personal taste. Stephen may have agreed that Blackstone's approach was narrow by emphasising an 'exclusively legal view of the national institutions'; however, he thought that Bentham's 'theory of the constitution, and by consequence of the history of England, is not only false, but has simply no relation to fact whatever' [63] (p. 27).

Although writing in a different period, even Stephen's text comes across as congratulatory—although he focuses on the common law aspect of the English legal system. Interestingly, Stephen wrote in a period during which, according to Bentley, the romantic style of history writing started to adopt a far more analytical approach, better known as political history [22] (pp. 68–99). Indeed, Stephen does not tend to exaggerate the superiority of the common law, although he does compare it with the (struggling) French legal system, thus indirectly emphasising its 'success'. Nevertheless, his text reveals a common view on the order of things, social hierarchy, and authority. In his observation about the importance of the separation of powers principle, Stephen argues that Parliament, as a legislative body separated from the ruler, 'represents directly the will of a large proportion of the community' and therefore 'it is unnecessary to distinguish between the morality of the legislator and that of the persons legislated for', because 'the two may be considered as practically identical' [9] (p. 77). In Stephen's time, 'the proportion of the community', that is, the voters, only included male property owners; this in itself does not mean that his factual account was inaccurate or generated a distortion of history. Rather, it is his conviction as to the legitimacy, fairness, and unproblematic nature of this status quo which can be contrasted with what appeared to be a somewhat fragmented society. Certainly, evidence suggesting popular resistance to the narrowness of voting rights is reflected in the passing of three related statutes: the Representation of the People Act 1832 (2 and 3 Will.IV), the Representation of the People Act 1867 (30 and 31 Vict. c. 102), and the Representation of the People Act 1884 (48 and 49 Vict. c. 3).

Despite criticising the *Commentaries* for its 'overstrained praise', 'the fact still remains', Stephen said, 'that Blackstone first rescued the law of England from chaos' [9] (p. 214). However, he was far more ambitious than Blackstone; he seriously took the notion of having 'a collection of general principles' (what were termed 'Codes' elsewhere) [43] and presented a Criminal Law Code of over two hundred pages of indictable offences to Parliament in 1878 [69]. Although he considered it a work of art, he was concerned with Parliament's overcautiousness: 'I am disposed to think that the difficulties of codifying the law are for the present practically insuperable' [70] (p. 362). Indeed, the Bill never passed, and no codification of English law ever followed. However, Stephen continued to express his concern as to the 'extreme obscurity and intricacy' of the law—for him, 'the vast masses of matter contained in our law libraries' needed to be rearranged and condensed. He felt that this would not only benefit the 'administration of justice', but that 'it would enable Parliament to legislate on legal subjects with its eyes open' [67] (pp. 200–201).

### 5. The Study, Practise, and Reform of the Law

The coincidence of the eighteenth century's scientific turn and the rising profile of university studies in England might not necessarily have been accidental. Berman explains that the 'legal science' community was aware of a 'collective responsibility' to generate, share, and train future generations in new 'scientific [legal] knowledge' [42] (p. 937). Whilst the notion of university as a collective (and legal) enterprise was more established in countries such as Italy and Germany, it is from the dissatisfaction with the Inns of Court, typical of the English legal system, that university legal education acquired momentum in England [71] (p. 472); [65] (pp. 731, 733) (albeit not without controversy and resistance [58]). Blackstone set in motion the debate and drive for university legal education [65] (p. 274), but it was not until the second half of the twentieth century that university education became the predominant route for aspiring legal professionals. In 1846, the Select Committee on Legal Education criticised the lack of commitment to the 'scientific character' of the law, and it was only at University College London (UCL) that an attempt was made to actually teach English law [71] (p. 473). In a speech at the House of Commons in 1871, Sir Roundell Palmer asserted that 'it is no untrue description of our law studies to say that they have been unscientific, unsystematic, desultory, and empirical.' [72] (p. 4); and, in an address to the American Bar Association in 1895, Thayer intimated that in England, 'the systematic teaching of law in schools is but faintly developed' [73] (p. 4). In 1877, Stephen went as far as suggesting that 'laws' should be the 'subject of liberal education', and that if 'the law were thrown into an intelligible shape [ . . . ] (it) would constitute a new branch of literature and public education' [67] (p. 362).

Butterfield's argument that Whig histories were written as a 'handy rule of thumb' for present purposes [25] (pp. 11,17) might in fact be a valid assertion, and it does not have to be taken in the negative. For Mayr, Butterfield's critique of Whig histories missed the point of what these histories meant to do [74] (p. 305). Indeed, Prest explains that 'the Commentaries do not purport to be an historical treatise' [75] (p. 192). Both Blackstone and Stephen embarked on the project of writing a text on criminal law because of their dissatisfaction with the chaotic literal presentation of the current penal system; according to them, this affected the way the discipline was studied and how the law was applied in court. It is possible that by fulfilling public roles after going through their own academic experience, Blackstone and Stephen were in a prime position to assess this gap in legal literature and education. Indeed, the writing of a history manual for the study of law was not coincidental. Arnold explained that history was meant to be relevant to concerns faced by current society [15] (p. 46); whilst Sunderland went as far as suggesting that 'it may be said of England far more than of any other European nation of today that her laws are the record of her life' [76] (p. 572). Lord Bolingbroke, writing in 1779, emphasised the importance of history as a 'philosophy teaching by examples [ . . . ] for our own time' [77] (p. 14). Blackstone's writing fits well within this style of philosophic history; according to Trevor-Roper, this style was distinguished by the emphasis it placed on the idea of 'the organic nature of society' and its progress [78] (p. 4). Smith suggests that Blackstone's aim was to demonstrate how criminal law had evolved 'through the combined forces of "love and liberty"' by emphasising the importance of the interactive nature of law and morals [79] (p. 24). In other words, the study of law within its historical context was a well-established practise driven by the idea that there was a 'close and intimate correlation [ . . . ] between the progress of English law and the moral, social, and industrial development of the people.' [75] (p. 572); [80] (p. 4). For Cross, the history of law 'should prove of incalculable assistance to the student of the law of today', not least because it might clarify 'many obscurities' of modern law; Cross went on to accuse 'the smug philistinism who cares not to look behind, away from, or beyond the circle of its present interest' of showing a lack of sympathy when dealing with legal problems [26] (pp. 654, 656).

Although Blackstone specifically targeted the study of the law, and Stephen was more interested in creating a companion to be adopted for the purpose of law reform, both were concerned with the way the law was practised. In fact, only students at Oxford and

Cambridge universities could benefit from a systematic study experience, although the study there was confined to the remits of Roman law (According to Stein, Oxford and Cambridge 'never accepted the common law as worthy of university study' [81] (p. 435). Otherwise, the English lawyer-to-be was expected to go through a self-didactic experience in the professionally focused (especially to counteract Oxford and Cambridge) Inns of Court [65] (p. 731); [82] (p. 419). Reflecting upon his student life, Lord Bowen recalled the endless Acts of Parliament, the cases and the authorities, the piles of forms and of precedents calculated to extinguish all desire of knowledge even in the most thirsty soul (Quoted in [65] (p. 733).

The importance of studying the law within its historical context was clearly expressed by Blackstone in his first chapter. Despite recognising that Parliament is the 'guardian(s) of the English constitution' [8] (p. 9), his emotions were clearly expressed when discussing the ignorance of Members of Parliament: 'how unbecoming must it appear in a member of the legislature to vote for a new law, who is utterly ignorant of the old! What kind of interpretation can he be able to give, who is stranger to the text upon which he comments!'. He went as far as saying that while there was no profession which did not require some extent of study, the 'man of superior fortune', that is, the Member of Parliament, 'thinks himself *born* a legislator' [8] [italic in origin]. Similarly, Blackstone considered the judiciary to be the backbone of the common law system, but here he also warned against inadequate knowledge; he suggested that 'without any skill in the laws he [the judge] will boldly venture to decide a question, upon which the welfare and subsistence of whole families may depend! [ ... ] and where, if he chances to judge wrong, he does an injury of the most alarming nature' [8] (p. 12). This concern had not died out by the following century; Stephen emphasised the importance of the systematisation of the law so as to avoid the situation where 'men acquire a vast amount of ill-arranged knowledge' [67] (p. 214).

However, the relation between the study of law and its application was of greater concern than either Blackstone or Stephen made it out to be. Writing in 1722, Thomas Wood explained that there had been a reluctance to 'bring the laws of England into a method'; Wood talked about the urgent need for a methodical study of the law, emphasising that the current state of legislation and common law needed to be reassessed in order to facilitate that [83]. As mentioned earlier, despite the congratulatory style of the historical narrative adopted by Blackstone and Stephen, and Bentham's conviction that the government of England was 'the finest and most excellent of any of the world ever yet saw' (Cited in [84] (p. 353), reality may have been less encouraging. A review by Burke of the *Commentaries* stated that 'we must owe no trivial obligation to any gentleman [ ... ] who will take the pains to remove any part of the obscurity in which our system of law is involved'; accordingly, Blackstone 'has entirely cleared the law of England from the rubbish in which it was buried' (Burke 1767, cited in [85] (p. 286). However, little did Burke know that the need for reform was far more acute. Indeed, during this period, the justice system was experiencing public distrust due to what seemed to be a discretionary, disproportionate, and at times inconsistent application of the law. There was particular concern about criminal offences, given the greater impact this had on the victim, the offender, and society at large (See related discussion in [86] (pp. 31–46). Blackstone, for instance, was at the forefront of projects attempting to reform the criminal justice system and penal law; capital punishment was being revisited, and transportation, though considered to be ideal, was embargoed by the colonies [86] (pp. 46–57). Stephen also encouraged law reform in the areas of criminal offence and the criminal trial. In writings such as 'Suggestions as to the Reform of the Criminal Law', he proposed revisiting the law on summary convictions, the arrangements for holding assizes and quarter sessions, the question as to the interrogation of the accused, and the question of appeals in criminal cases [87] (p. 738).

Moreover, throughout Blackstone's and Stephen's lives, the British Empire was routinely challenged both at home and abroad. As an example, White's research on popular politics in the eighteenth century illustrates the influence of public opinion as expressed through popular media such as satirical cartoons, pamphlets, and newspapers; in these,

governmental elections were ridiculed, politicians were shamed, and colonial rule was criticized. Newspapers reported events such as riots and street protests, which were frequently accompanied by army interventions [88]. Hogarth's work, although exaggerated, was meant to reflect the debauched nature of government and politics, as depicted in his 1755 *An Election Entertainment*; this satirical print is one of four in Hogarth's series *The Humours of an Election* 1755–1758, aiming to illustrate the corruption of Whigs and Tories during the election (The paintings and a description are available in Graphic Arts [89]). In 1788 the public could gather in front of print shops to be amused by prints such as *The Bow to the Throne, alias the begging bowl* by the caricaturist James Gillray, criticizing the wealth made from colonial rule in India [88]. Another caricature by Gillray was released in 1805 depicting Pitt and Napoleon 'eating' the globe (image available in [90]). The judiciary was not spared from satire either when Hogarth's collection *The Bench* was advertised by the *Whitehall Evening Post* (London Intelligencer) in 1758 [91]. Furthermore, governmental weaknesses were picked up by political agitators such as John Wilkes (Who himself was also Hogarth's object of satire, see explanation in Art of the Print), a Member of Parliament who vivaciously criticised English politics during the 1760s in the political newspaper *North Briton*; in 1762, Wilkes wrote that 'fiction and falsehood are the two main pillars of [the] political state' [92]. In an article for *The London Review* in 1835, James Mill wrote that 'as things are managed in England, that protection upon which [ . . . ] all the happiness of society depends—is most imperfectly afforded.' [93] (James Mill (Stuart Mill's father) signed some of his articles as P.Q.). In response to Parliament's reluctance to pass Stephen's Penal Code Bill in 1878, an article in the *Saturday Review* complained that 'it is plain that we are misgoverned in respect of one of the most important departments of the Executive [ . . . ]; and that the Government will be justly chargeable with apathetic disregard of the public welfare [ . . . ].' [94] (pp. 558, 559).

Likewise, major threats to governmental stability were endured from the American front (Riots started to erupt in 1765 with the imposition of extra taxes by the British government; the situation escalated with the 1773 Boston Tea Party, and then the American War of Independence in 1775 [95]), from Spain and France (This included the War of Jenkins's Ear and the War of the Austrian Succession (broadly 1740–1748), the Seven Years' War with France (1756–1763), and the French Revolution starting in 1789 [95]); these further affected the English public's dissatisfaction with Parliament. For instance, inspired by the drivers of the French Revolution, the London Corresponding Society called for universal male suffrage; the Hampden clubs advocated parliamentary reform; in 1817, working class men marched from Manchester, aiming to reach London before being dispersed by the army. A couple of years later, a demonstration against high food prices and for parliamentary reform ended up being known as the Peterloo massacre. And another failed attempt at parliamentary reform in 1830 led to violent riots in Bristol, Nottingham, and Derby (The society was banned by the government in 1794, as were the trade unions in 1799–1800; the Hampden were banned in 1818. In Peterloo, the army's attempt to disperse the crowd led to eleven deaths and hundreds were injured [95]). During the late 1830s and the early 1840s, Parliament faced several petitions by the People's Chartists demanding social and political reform; likewise, the women's suffrage campaign was slowly gaining momentum from the 1860s [96]; and as mentioned earlier, several Acts of Parliament were passed throughout the century to reform the law on enfranchisement.

## 6. Conclusions

The aim of this paper was to explore how the doctrine of criminal law and its teaching became positivist in nature. Legal scholars have suggested that the abstraction of the doctrine of criminal law and the alleged disguised tension between the state and the citizen have much to do with the way the law was categorised by early scholars such as William Blackstone and J.F. Stephen. There is no doubt that these writings brought about important and significant implications on the teaching and study of the doctrine of criminal law; however, given that Blackstone and Stephen reflected a political independence of thought,

it becomes difficult to surmise that positivism was an intentional master plan of autocratic social control, as is sometimes argued. This external legal history suggests that Blackstone and Stephen wrote their texts by following the literal style of their period. The fact that the writing style reflected a scientific approach to the 'creation' of knowledge meant that several disciplines, including law, were written and taught in a positivist fashion. Blackstone's and Stephen's positivist legal histories provided, during their time, a novel approach to the study and teaching of law. They also played a huge role in facilitating efficient practise and legal reforms.

**Funding:** This research received no external funding.

**Institutional Review Board Statement:** Not applicable.

**Informed Consent Statement:** Not applicable.

**Data Availability Statement:** Not applicable.

**Conflicts of Interest:** The authors declare no conflict of interest.

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
