# Peer review of "How to Write a Positivist Legal History: Lessons from the 18th and 19th Centuries English Jurists William Blackstone and James Fitzjames Stephen"

_2409-9252, doi:10.3390/histories1030017_

Round 1
Reviewer 1 Report
I recommend the author's paper for publication. The arguments flow logically, and the paper is generally well-structured. The content of the paper is original and of high quality. Among other things, the paper presents a compelling argument that the conceptual organization of the criminal law of England, as undertaken by Blackstone and Stephen, was influenced by a widespread interest in the scientific method during their lifetimes, and this led to the law being taught and practiced as if legal positivism were true. There are some typographical errors that need to be corrected. For example, on page 3, "lawyer" is spelt "layer".
Author Response
Thank you for your valuable comments.
I have spell checked the work, and I have sent it to a specialised service for this purpose too.
Kind regards
Reviewer 2 Report
This is a well-considered and well-argued text, which addresses an issue relevant to the history of criminal law. I believe it can be published in its present form, but it would benefit from further reflection and reference to several theoretical and methodological issues.
The main one has to do with the fact that the author adopts an approach centred on the intentions of the authors, but does not offer references or a discussion of the issue, which has a long history of debate following the approach by Quentin Skinner. Something similar happens with the issue of codification, which is in the background of the whole text: the work assumes too much of an English perspective on the rise of positivism in criminal law, but does not take into account that the trend towards codification was occurring in other neighbouring countries, as Bartolomé Clavero, among many others, has studied; a minimal comparative treatment would have been illustrative here.
The author's treatment of contextualisation also deserves comment: the contexts of Blackstone's and Stephen's works are separated by a century, but the author tends to put the analogies before the differences, which were rather more marked than he suggests in relation to the status of scientific discourse in general, and in the field of constitutional law and political theory in particular; a greater emphasis on the cumulative nature of scientific discourse between one and the other temporal context would rid the text of the sensation of talking about two interchangeable authors. In that sense, the author misses the opportunity to reflect on the extent to which the two authors form a tradition. The literature he offers on the scientific revolution and its aftermath is not always very up-to-date.
Finally, there is the question of the relationship between praxis as judges and the elaboration of historical narratives, a field of reflection that has some now classic texts such as that of Carlo Ginzburg that could be brought up; likewise the reflections on Western legal culture and the construction of the history of law, by Antonio Manuel Hespanha, if only once again to specify the English trajectory of common law.
Author Response
I found the comments made by referee 2 valuable, and following the editor’s advice that these might elevate the quality of the writing- I have indeed taken these on board as follows:
I have injected some theoretical considerations drawing on Skinner, Ginzburg and Hespanha. I feel that this addition has added academic rigor to the writing.
I recognize that the context of the writing is narrow by merely focusing n the English Legal system; yet, as advised I added a paragraph explaining that the scientific trend and its main consequence (codification) was a tradition happening not only in England- and I feel that this adds reliability to the writing.
An important point made was the need to justify the examination of two different literal styles spanning across two centuries; I think, as suggested, the discussion was made weak by referring to Blackstone and Steven almost as reflecting the same approach to the writing of history. Hence, tweaking the narrative in such a way to suggest that they were part of the same ‘tradition’ of writing and in a cumulative way (also by adding the codification aspect), was in fact very helpful.
Reviewer 3 Report
The paper’s general (and worthy) objective seems to be a re-evaluation of Blackstone’s Commentaries and Stephen’s History of the Criminal Law.
In the author’s words (at p.3), ‘the aim is to explore why the teaching of criminal law took the form of a positivist enterprise: that is, law which has been posited (laid down) by the legal system of the country. This is in contradiction to natural law theory, for example, which recognises the legitimacy of a law in ways which are more profound than the mere cohesive power of the state’.
To claim that Stephen’s History was a product of the positivist method is quite orthodox, insofar as Stephen himself professed as much, starting his definition of criminal law [ch. 1 bk. 1] with statement that law is distinguished from morality and is coercive (and broadly adhered to Austin’s definition of law). There is a no doubt serious critical literature that downplayed Blackstone’s commitment to the natural law theory (see, for example, M. Lobban, ‘Blackstone and the Science of Law’, The Historical Journal , Vol. 30, No. 2 (Jun., 1987), pp. 311-335), but the relevant arguments are not mentioned in the paper. The author seems to take the classification of Blackstone as a positivist as an unquestionable fact. However, Blackstone’s own statements hardly support this classification of him as a positivist and a critic of the natural law theory:
Upon these two foundations, the law of nature and the law of revelation, depend all human laws; that is to say, no human laws should be suffered to contradict these. There are, it is true a great number of indifferent points, in which both the divine law and the natural [law] leave a man at his own liberty; but which are found necessary for the benefit of society to be restrained within certain limits. And herein it is that human laws have their greatest force and efficacy; for, with regard to such points as are not indifferent, human laws are only declaratory of, and act in subordination to, the former. To instance in the case of murder; this is expressly forbidden by the divine, and demonstrably by the natural law; and from these prohibitions arises the true unlawfulness of this crime (Blackstone’s Commentaries, Introduction, Sec. 2 [italics added]).
At p. 14, the author claims: “And it is within this context that Blackstone recognises the importance of moving away from a law which is merely natural, because the laws ‘denote the rules, not of action in general, but of human action or conduct.’ Therefore, because ‘man was formed for society’, these rules have to be ‘permanent, uniform, and universal’” [italics added]. If the reference to ‘a law which is merely natural’ meant to contrast the natural law theory with the positivism, this is a misinterpretation. The whole quote is as following:
This then is the general signification of law, a rule of action dictated by some superior being: and, in those creatures that have neither the power to think, nor to will, such laws must be invariably obeyed, so long as the creature itself subsists, for its existence depends on that obedience. But laws, in their more confined sense, and in which it is our present business to consider them, denote the rules, not of action in general, but of human action or conduct: that is, the precepts by which man, the noblest of all sublunary beings, a creature endowed with both reason and freewill, is commanded to make use of those faculties in the general regulation of his behavior.
Moreover, Blackstone’s reference to ‘man was formed for society’ is a part of his discussion of the formation of the law of nations [the basic concept of the natural law theory, alongside of ‘the law of nature’ & ‘the state of nature’] that ends up with its classical definition: quod naturalis ratio inter omnes homines constituit, vocatur jus gentium [that rule which natural reason has dictated to all men, is called the law of nations.] The last quote on ‘permanent, uniform, and universal’ rules, is, indeed, linked to civil law, defined by Blackstone as municipal law [the only kind of law that positivism ‘proper’ would focused on], referring to the requirement of generality of law [sort of anticipating Fuller’s 20th century vision of the natural law theory- Fuller’s inner morality of law].
The author seems to implicitly equate positivism and ‘”scientific knowledge” approach’ to law’. But there is only most general discussion of such approach, not particularly well structured, and even contradictory/inconsistent at some points. For example, the above mentioned ‘scientific’ approach is linked to ‘systematisation’ at pp. 8 & 11, but at p.12 the reference is made to the scientific method that is now described as ‘unsystematic’. Then at p.13, in a new turn (without a clear relevance to the issue at hand) the author claims that ‘[t]he scientific trend … represented a way of life’. At p. 14 , the author states as a given fact that ‘Blackstone and Stephen set forth to… create a systematic user-friendly guide, in a period where rules of science acquired a predominant importance in the study and writing of any discipline’. One may expect at least some elucidation of this ‘scientific’ approach in application to the texts of Blackstone and Stephen.
At p. 15 the author seems to [rightly] imply that for Stephen’s ‘interests of morality’ have nothing to do with application of law. But next, the author claims that [positivism?] encouraged the fiction that the development …of the law had been driven …‘by the agency of actual legislation working in obedience to the higher moral aspiration of growing societies’ [italics added]. So would positivist scientific approach be insulated from morality or not? At p.21 the author mentions [approvingly] Smith’s suggestion “that Blackstone’s aim was to demonstrate how criminal law had evolved ‘through the combined forces of “love and liberty”’ by emphasising the importance of the interactive nature of law and morals’”.
Perhaps, placing Blackstone within the ‘Whig histories’ became more common in the recent decades. Still Blackstone’s depiction (at p. 4) as ‘a Tory sympathiser within a Whig government’ [italics added] needs at least to be clarified ( insofar as he had been standing in election as a Tory). The reference to this statement is to the respectful secondary source, but the footnote [14] lacks page referencing (!) (by the way, footnotes 15 & 18 also lack page references). The author noted (at p. 16) one rationale for classifying Blackstone’s Commentaries as belonging to ‘congratulatory histories’. Blackstone had prised ‘the separation of powers principle’. Perhaps, the term is a bit of misnomer [due to the modern connotation of it as relating to legislative, executive and judicial powers] , since Blackstone’s reference was ‘to the king; secondly, the lords spiritual and temporal, which is an aristocratical assembly of persons… and thirdly, the house of commons, freely chosen by the people from among themselves’). In place of the discussion of the parliamentary sovereignty at pp 17-19, one may expect some evidence of ‘congratulatory histories’ in application to criminal law specifically, due to the asserted focus on criminal law. At the second part of p. 18, the author at last turns to the issue of differences in approaches to criminal law between Blackstone & Stephen, but merely passingly (just mentioning that Stephen was ‘far more ambitious than Blackstone; he took seriously the notion of having “”a collection of general principles”’, without any further elucidation).
At p. 21, the author sensibly argues [as a rebuke of some criticism of [the so called] ‘Whig histories’] that ‘both Blackstone and Stephen embarked on the project of writing a text on criminal law because of their dissatisfaction with the chaotic literal presentation of the current penal system; according to them, this affected the way the discipline was studied and the law applied in court’. Here, again, one may expect some examples of systematisation in the texts at hand.
In sum it is not clear what is the paper specific focus: is it on systematisation of criminal law on the example of Blackstone & Stephen texts? The conclusion (p. 26) states that ‘[t]he aim of this paper was to explore how the rationalisation and systematisation of the doctrine of criminal law and its teaching became positivist in nature’. The paper barely mentions any doctrines of criminal law. Moreover, there are so many divergences from the issue of systematisation of criminal law. At p. 23, for example, the discussion moves to legislation and judges. At p.24, the focus shifts to the proposals for criminal reforms (in Blackstone’s case without direct reference to his work/s or any specific details). It may indeed make sense to textually compare the respective plans for reforms against the background of the respective criminal law frameworks of Blackstone & Stephen, but the paper ‘moves on’. Immediately after (at p. 24), the author turns to criticism of the British empire and the need for the parliamentary reforms in the 19th c.
Not only the focus of the paper is hard to pin down but also the actual/specific argument. In the last section (p. 27), the author notes that ‘these writings brought about important and difficult implications on the teaching and study of the doctrine of criminal law’. This statement seems to relate to the declared aim of the paper at p. 26 (referred to in the paragraph above). But the paper has not elaborated what are ‘important and difficult implications’. The paper states, as if in conclusion, that ‘Blackstone and Stephen wrote their texts by following the literal style of their period… the writing style reflected a scientific approach to the ‘creation’ of knowledge’. Notably, the statement that the texts at hand reflected a certain scientific approach has been taken (at p.14) as a given premise [without actually elucidating it in application to these texts]. Hence, overall reasoning seems to be circular.
Author Response
I am grateful to the comments made by reviewer 3, however, from its opening sentence, I feel that perhaps the aim of my writing was missed. I can see that the misinterpretation of my writing continues in the comments that follow. I would have agreed with these comments if my writing reflected the aim suggested by the reviewer- but this is not the case. I appreciate the reviewer knowledge of the two sources, but the comments concentrate too much on their own approach to a possible examination than on my own- and this is disconcerting.
Round 2
Reviewer 3 Report
-